# The Life Impact Burn Recovery Evaluation (LIBRE) Profile: Historical Overview and Future Directions

**DOI:** 10.3390/ebj6020023

**Published:** 2025-05-14

**Authors:** Colleen M. Ryan, Jeffrey C. Schneider, Pengsheng Ni, Mary D. Slavin, Amy Acton, Ananya Vasudevan, Allan Sosa-Ebert, Lewis E. Kazis

**Affiliations:** 1Department of Surgery, Massachusetts General Hospital, Harvard Medical School, Boston, MA 02114, USA; 2Shriners Children’s Boston, Boston, MA 02114, USA; 3Department of Physical Medicine and Rehabilitation, Schoen Adams Research Institute, Rehabilitation Outcomes Center (ROC), Spaulding Rehabilitation Hospital, Harvard Medical School, Boston, MA 02129, USA; jcschneider@mgh.harvard.edu (J.C.S.); mslavin@bu.edu (M.D.S.); lek@bu.edu (L.E.K.); 4Department of Health Law, Policy, and Management, Boston University School of Public Health, Boston, MA 02118, USA; psni@bu.edu; 5Phoenix Society for Burn Survivors, Kentwood, MI 49508, USA; 6Department of Anesthesiology, Tufts Medical Center, Boston, MA 02111, USA; avasudev@bu.edu; 7Elmhurst Hospital Center—Internal Medicine Residency Program, Elmhurst, New York, NY 11373, USA

**Keywords:** burn, thermal injury, PROM, patient reported outcome measure, LIBRE, life impact burn recovery evaluation, long-term outcome, rehabilitation, reintegration, community participation

## Abstract

The Life Impact Burn Recovery Evaluation (LIBRE) Profile was developed to assess long-term social participation outcomes for adult burn survivors. Traditional clinical burn recovery outcomes focus on early physical complications and psychosocial issues, but there is a growing need for quantitative measures of long-term recovery that assess experiences deemed relevant to burn survivors. The LIBRE Profile, co-produced with input from burn survivors and clinicians and grounded in the World Health Organization’s International Classification of Functioning, Disability and Health (WHO-ICF) conceptual framework, addresses the measurement gap by focusing on six domains of social participation: social interactions, social activities, family and friends, work and employment, romantic relationships, and sexual relationships. The LIBRE Profile uses Item Response Theory (IRT) and computer adaptive tests (CAT) to minimize respondent burden while maintaining accuracy. Psychometric evaluations have validated the LIBRE Profile as a reliable and clinically useful tool that can help clinicians and burn survivors monitor recovery and inform personalized care. Future work includes LIBRE Profile development for pediatric populations, further international language translations, and the development of an APP for broader personal and clinical use. This paper provides a comprehensive overview of the LIBRE Profile’s development, psychometric foundations, and future directions, advocating for its adoption in clinical practice and burn survivor communities.

## 1. Introduction

The purpose of this LIBRE “Capstone” paper is to summarize and integrate the first 10 years of the development and work on the adult LIBRE instrument and to make the LIBRE Profile easier for people to use and understand. The dissemination of the instrument through this paper provides a central place to find information on the instrument and present a summary of each paper so that researchers can quickly direct any specific questions regarding its development, reliability, and validity to an original literature source. Additionally, assembling the body of work in one place demonstrates how the process of scientific co-production that includes the patients, clinicians, and scientists is accomplished. This project began from identification of a problem, e.g., the lack of knowledge of the long-term health needs of burn survivors, which was brought to an advisory group by burn survivors and clinicians. This problem was addressed using co-production from grant writing through product development and continues through implementation. Inclusion of the stakeholders in the development of the product improves the uptake of the work.

Traditional clinical outcomes measured after burn injury include early post-burn mortality, use of in-hospital resources, such as ICUs and ventilators, the occurrence of specific physical complications, such as infections, contractures or heterotopic ossification, and the epidemiology of psychological issues, such as PTSD, depression, or anxiety [1,2,3]. Quantitative measurement of early outcomes after burn injury is important for allocation of resources and optimization of acute and early post-burn care [4]. The measurement of the long-term course of burn recovery from the patient viewpoint is a nascent field that is less defined, more conceptual in nature, and often based on qualitative work [5]. A more recent PROM specific to burns, for example, is the CARe Burn Scale, published in 2018 [6]. A quantitative measurement of long-term burn recovery from the burn survivor’s perspective would be useful to identify the appropriate allocation of resources, compare and optimize interventions, identify specific phenotypes to assist in clinical decision-making, and predict trajectories at the individual level to promote assessment of progress in recovery from burn injury [7]. While some legacy patient-reported outcome measures begin to address these issues, their scientific underpinnings could be improved [8]. Additionally, barriers to implementation of tools in clinical settings include technical issues, respondent burden, lack of uniform scoring guidance, score interpretation, and specific recommendations for clinical interventions among those with scores below established norms [9].

The World Health Organization’s International Classification of Functioning Disability and Health (WHO-ICF) provides a conceptual framework for the assessment of long-term disabilities using three overarching domains: body function and structural impairment, activity limitations, and participation restrictions [10]. These domains interact and are influenced by the health condition and by environmental and personal factors that influence an individual’s ability to meaningfully engage in daily societal activities at home and in the greater community [11]. Informal discussions with burn survivors often suggest that a measure to assess social participation would be useful in understanding the recovery process after burn injury [12]. However, this concept requires a granular examination of the specific aspects of social participation that are deemed relevant to the burn-injured population. To be clinically meaningful and address issues that are most important to individuals with burn injuries, a metric developed to assess social participation must be viewed from the lens of the burn survivor as well as clinician specialist caring for this population.

Generic patient reported outcome measures exhibit both strengths and limitations regarding assessing outcomes of persons living with burn injuries. The Patient Reported Outcome Measures Information System (PROMIS) includes scales that assess satisfaction with social roles; however, items were not developed or tested in populations that include burn survivors [13]. While some PROMIS items may be relevant for burn survivors, the PROMIS measures lack the granularity required to fully describe the burn survivor’s unique experience in social situations to provide a useful clinical/social tool. On the other hand, the generic nature of the PROMIS scales allows comparison of the measure scores across different sociodemographic and clinical conditions, leveraging the collective knowledge across different clinical conditions. Ideally, generic and condition-specific PROMs can be bridged to provide a wider understanding of impact on the life of burn survivors. Items for the original item pool for the LIBRE Profile to be described were both generic and condition-specific for burns. Based upon cognitive interviews and focus groups with burn survivors, generic items were modified so that they were condition-specific.

The National Institutes on Disability, Independent Living and Rehabilitation Research (NIDILRR) previously funded the development of a condition-specific metric in 2014 to evaluate social participation following burn injury in adults 18 years and older. Key aspects of the project include: use of the WHO ICF framework, co-production with burn survivors and clinical experts, and the use of Item Response Theory (IRT) to develop a computer adaptive test (CAT) to minimize respondent burden while maintaining accuracy of the metrics [14,15]. A highly rigorous approach resulted in development of the Life Impact Burn Recovery Evaluation (LIBRE) Profile for adult burn survivors 18 years of age and older. Efforts to promote widespread dissemination to the burn community include assignment of clinical meaning to numeric scores and the involvement of implementation scientists to outline effective dissemination strategies. Collaboration with the Shriners Hospitals for Children in the development of pediatric CATs for the measurement of burn recovery in preschool, school age children, and teenagers also has been developed [16]. Through the creation and use of the pediatric and adult LIBRE Profiles, clinicians and researchers have begun to quantitatively and qualitatively understand how burn injuries influence an individual’s ability to meaningfully participate in their daily life activities [11,16]. In this paper, we summarize the LIBRE project findings to lay the groundwork for future studies and promote greater use of the instruments in the clinical and burn survivor communities. We provide a figure summarizing the development of the LIBRE Profile and its current and future applications (Figure 1).

We begin with the framework for the Adult Version of the LIBRE Profile, historically the foundation for the LIBRE program, followed by the psychometric work that paved the way for development of the Computer Adaptive Test (CAT).

## 2. Conceptual Framework Development

The work done to develop the Adult LIBRE Profile established the domains influencing social participation within a conceptual framework rooted in the WHO ICF. The WHO ICF is an internationally recognized model of health that seeks to integrate and display the relationship between biological and psychosocial functioning as it relates to overall wellbeing [17]. To better understand the role social participation plays in meaningful life experiences after burn injury, Kazis et al. utilized grounded theory methodology to develop a conceptual framework that aims to understand the role between the different domains of burn survivors’ lives that are influenced by their injuries [1,18,19]. The conceptual framework began with focus groups of burn survivors with lived experiences to identify important areas of recovery. The initial framework identified social participation defined by two larger concepts, societal role and personal relationships. Further subgroups were chosen to identify key areas in which item development would occur [2]. Through the development of the LIBRE Profile, investigators identified the role of six key domains—social interactions, social activities, family and friends, work and employment, sexual relationships, and romantic relationships—that were deemed appropriate markers of social functioning. Calibration analysis of pre-existing legacy questionnaire items demonstrated the validity of the LIBRE Profile domains for use as a computer adaptive test (CAT). This work revealed that this conceptual framework appropriately showcases the importance of the selected domains [20,21]. By developing a holistic model, investigators highlighted the interaction between these domains and their influence on the psychosocial wellbeing of burn survivors [22].

## 3. Focus Groups and Development of Item Bank

Guided by the WHO ICF, the LIBRE sought to comprehensively understand and develop an assessment for social participation among burn survivors. A meticulous approach was adopted in order to develop the foundation necessary to examine social participation. This process began with an extensive literature review and consultations with six clinicians specializing in burn care, as well as direct input from burn survivors. The WHO ICF played a pivotal role in establishing relevant topic areas, subsequently shaping the interview guide utilized in focus groups with burn survivors and clinicians. Focus groups were conducted with clinicians and burn survivors segregated by gender, as sensitive topics provided greater opportunity for open discussions broken out by gender, with leaders of each group also gender-concordant.

Focus groups consisted of a total of 23 burn survivors, 11 females and 12 males with ages ranging from 20 to 66. The total time elapsed since burn injury spanned 6 months to 35 years, with total body surface area (TBSA) percentages ranging from 18% to 94%. In parallel, focus groups were conducted with 27 clinicians comprising 12 physicians including representatives from burn surgery, pediatrics, psychiatry, and physiatry and 15 other clinical professionals from other fields including nurse specialists in burn care, and representatives from several therapy divisions, psychology, social services, and child life. Years of experience ranged from 1.5 to 40 years across these clinicians. The focus group data were meticulously analyzed using grounded theory methods, employing inductive reasoning to construct hypotheses and theories. Based on extensive literature reviews, consultations with clinicians specializing in burns, and focus groups noted above with direct input from burn survivors, 192 items were identified for inclusion in the field-testing survey. This method facilitated the identification of two major domains—societal role and personal relationships—within the primary construct of social participation. Subsequent cognitive interviews with 23 burn survivors were conducted to assess the clarity and perception of 192 question items spanning six key domains ranging from work to informal relationships. Each question was presented with Likert scale responses, analyzed, and iterated upon based on valuable feedback from burn survivors.

## 4. Domains

The six domains related to social participation for those with burn injuries are tied to community integration following hospital discharge. These domains emerged from extensive qualitative and psychometric analysis including comprehensive literature reviews used to develop extensive item banks for the CAT administration with the domains: social activities, social interactions, family and friends, work and employment, romantic relationships, and sexual relationships [23]. The following gives a description of each domain.

### 4.1. Social Activities (15 Items)

There are several different types of social activities, spanning one-on-one interactions and small get-togethers to organized community events and informal gatherings where each is important in establishing a healthy level of social engagement [24]. It is important that individuals have an element of social participation in order to lead fulfilling lives. While these activities vary from individual to individual, those recovering from burn injuries often struggle to return to pre-injury levels of social interaction. Individuals not engaged in social activities can begin to feel isolated, detached, and less motivated to engage in their rehabilitation process of recovery [25]. Items included in the LIBRE Profile Social Activities domain aim to quantify an individual’s ability to accomplish social activities and participate with others and document changes over time. Fifteen items assess an individual’s perception of their abilities and what they do and how they have changed during their recovery [26]. Understanding how an individual’s social activities within their everyday environment have changed is crucial to better understanding how they may benefit from available supports and resources as well as their desire to engage with their surroundings [27].

### 4.2. Social Interactions (25 Items)

Understanding the way in which burn survivors are able to engage in social interactions with those around them is important for understanding the influence their injuries have on their ability to continue to interact with their environment. Individuals benefit from social support and communication with family, friends, and strangers to engage meaningfully in their daily lives [28,29]. Recovering from severe injuries can be extremely taxing on both the body and the mind; fostering social relationships can help mitigate this stress. Social support is extremely important in aiding reintegration into society for those who are recovering from burn injuries. For burn survivors, it can often be challenging to return to pre-injury levels of social interaction due to difficulty participating in, and dealing with the anxiety surrounding, social situations [30]. The 25 items that compose the LIBRE Profile Social Interaction domain aim to assess an individual’s ability to interact with familiar situations. Monitoring scores after discharge can establish the need for resources and support to promote recovery in this domain. While social interactions can be further influenced beyond the scope of injury by specific sociodemographic and contextual characteristics, these items aim to understand the social characteristics that are critical to each individual’s recovery [31,32,33]. Understanding how social interactions influence an individuals’ rehabilitation process is often important to ensuring successful recovery and reintegration into society.

### 4.3. Family and Friends (24 Items)

Social relationships are key to understanding the impact of burn injuries on the inherent ties and bonds with family and friends. Items focus on the amount of support one gets from others, their comfort level with family and friends, and activities pursued [34]. Understanding the strength and richness of relationships with family, friends, and peers is important to assess, as their support is often well connected to several outcomes such as life satisfaction, self-esteem, and participation in social and recreational activities [26].

Family and support are positive factors in the recovery process, no matter the level of severity of the burn injury [35]. The lack of family support is significantly related to development of mental distress in those with burns. Family support predicts a positive recovery [30]. Overall, the literature suggests that family support gives burn survivors the strength to endure the challenges and promotes resilience during the rehabilitation process, with an emphasis on long-term social adjustments. Items for this domain assess a burn survivor’s reintegration with family and friends, including the support the burn survivor desires from them, the need to avoid some family members, disliking how family acts around the survivor, and a range of other items involving family interactions with the survivor and their friends, assistance and comfort the survivor derives from family, communications with the family, and friends that are physically close [33,36]. As mentioned, first-hand by survivors, they wish their family had given them more time to process their feelings and grieve their burns [30]. It is important to assess these aspects of a survivor’s life to better understand their recovery and reintegration into the community.

### 4.4. Work and Employment (19 Items)

Work and employment often make up a significant portion of individuals’ daily lives, interactions, and responsibilities. Screening questions direct subjects to this domain for those currently working. Day-to-day activities are generally centered around work schedules, and individuals determine how to best find work–life balance. The work environment provides a unique component of social interaction as individuals engage with colleagues. Furthermore, workplace satisfaction and the ability to keep up with work requirements are different for each individual. The work by Verma et al. [37] indicates that return to work following a burn injury is related to social integration, while not doing so is related to ones’ physical impairment and quality of life. In those recovering from burn injuries, it can often be challenging to return to the workplace and maintain the same responsibilities due to a variety of reasons [37]. These reasons include, but are not limited to, changes in energy, motivation, fatigability, and physical consequences of disease [26,38]. Downstream consequences of challenges in the workplace can have significant impacts on an individual’s mental and physical wellbeing. Work and Employment domain items include content pertaining to work performance, responsibilities, energy, and absences secondary to the burn injury [39]. These questions aim to assess the individual’s status in the workplace as related to their functioning and the support of their workplace environment.

### 4.5. Romantic Relationships (28 Items)

Romantic relationships involve the specific association between two or, in some cases, more people, combining both the emotional and physical aspects of a relationship. This domain is often not addressed during provider–patient interactions where conversation may focus on the clinical aspects of the burn injury. Prior to completing the items, respondents confirm if they are currently in a romantic relationship. This domain covers content involving emotional closeness and physical attraction, sharing problems with each other, communication, and support [40]. Items include having someone to hug, hold, or kiss, someone to share what you dislike about yourself, someone to listen to your personal problems. Other items describe the positive and negative feelings toward your partner. Furthermore, this domain involves asking participants questions that allow clinicians to glean insight into how often they and their partner work on a project together or spend time together [34]. Romantic relationships are further assessed by asking about difficulty in telling your partner you love him/her, the comfort level with sharing your feelings about your burns, and how fun, loving, and happy the partner makes the survivor feel [33]. The perception of the partner accepting how the survivor looks is a critical part of continuing a romantic relationship after the injury. Assessing potential challenges for the burn survivor in romantic relationships is important, as they are likely to impact their overall wellbeing [30].

### 4.6. Sexual Relationships (15 Items)

Exploring the re-integration of burn survivors in their sexual relationships may not be addressed by the clinicians. In a United Kingdom survey of burn units by Hurley et al. [41], results are reported noting that sexual functioning is often not addressed by physicians in the clinical treatment of burn survivors. Burn survivors may experience changes in sexual behavior regardless of their relationship status. This domain is restricted to those burn survivors currently engaged in a sexual relationship. Sexual relationships are not necessarily specific to a particular person but involve the engagement and satisfaction with one’s current level and quality of sexual activity. Sexual relationships involve desire and interest toward sex, satisfaction with sexual activity, sexual confidence, and sexual intimacy [30]. The items in the Sexual Relationships domain are written in gender-neutral terms to be universal regardless of sexual orientation [34]. These questions cover topics including avoiding sexual contact due to burns, ability to engage in the sexual activities they like, interest in sex, emotional closeness during sexual activity, being found sexually attractive, being sexually excited, their partners’ enjoyment, satisfaction with frequency of sexual activity, and adding to the relationship. In terms of sexual functioning, psychological dysfunction and decreased self-esteem are linked to body image and disfigurement among those with burns [33]. A connection between severity of burn injury and sexual esteem, sexual depression, and sexual preoccupation is observed. Even though sexual function is highly relevant in terms of long-term quality of life, it is not often a priority for therapeutic intervention after burns [26,42], emphasizing the need to assess this domain in the target population [43].

## 5. Advanced Psychometrics and CAT Development of the Adult LIBRE Profile

### 5.1. What Is a CAT?

A computer adaptive test (CAT) is a dynamic assessment that tailors the difficulty of questions presented to the test-taker based on their previous responses, providing a method to obtain brief and precise trait estimates that can be incorporated into patient-reported outcome assessments. CATs allow for individual estimates at the item level to enhance efficiency and accuracy on a common metric [44]. The CAT is based on no more than 10 questions for a specific domain, with sufficient comparable precision for all items being included in the item pool battery. Unlike traditional fixed-form tests, a CAT adjusts the level of item difficulty in real time, optimizing the precision of the assessment by focusing on the individual’s ability level. This personalized approach enhances efficiency and accuracy, ensuring a more precise evaluation of the respondents’ abilities across various domains. Unlike classical test theory methods, Item Response Theory (IRT) models provide a rigorous approach for measuring the social impact of burn injuries on social participation.

### 5.2. Psychometric Evaluation—Calibration Study

Following the focus groups with burn survivors and clinicians to develop item pools, a study was conducted to calibrate items. The calibration process served a dual purpose, aiming to create, measure, and define the influence of burn injury survival on social participation and establish a baseline and standard metric for further evaluation [45]. This rigorous and meticulous procedure involved the systematic administration of 192 questions spanning various facets of social participation to a substantial, rich convenience sample of 601 burn survivors representing a range of clinical and sociodemographic characteristics. Inclusion criteria for participants were individuals aged 18 and above, with a minimum of 5% total body surface area burned (TBSA) and burns to critical areas including face, hands, feet, or genitalia. There were no exclusionary criteria for otherwise eligible participants with acute health or psychiatric problems, which enriched the sample heterogeneity. The study focused on participants from the United States and Canada who had the ability to read and understand English. Recruitment strategies were diverse, leveraging platforms such as the national Phoenix Society Organization for burn survivors, peer support networks, social media channels, and traditional mail. Eligibility screening was conducted through phone interactions, with demographic information, including age, sex, race, ethnicity, marital status, education level and time since burn injury, TBSA, and other aspects of the burn injury, systematically collected. Three filter questions were strategically incorporated, directing participants based on their current work status, romantic relationship status, and sexual activity. The survey itself was administered either over the phone by trained interviewers or through online platforms, ensuring a comprehensive and inclusive data collection process.

### 5.3. Psychometric Testing and Factor Analysis

In the refinement process for the questionnaire, items exhibiting over 90% response in one category and low overall frequency were systematically eliminated. The subsequent step involved an exploratory factor analysis (EFA), employing the full information maximum likelihood estimation method (FIMML) and the oblique Crawford–Ferguson (CF) quartimax rotation technique. Factor loading patterns were scrutinized, ratios were examined, and the percentage of variance was determined to extract 1–10 factors. The decision on the number of factors was informed by a thorough examination of results, considering the similarities between each factor’s content and its alignment with the conceptual framework. Following the EFA, a confirmatory factor analysis (CFA) model was employed to assess the extent of item factor loadings. The calibrated items emerging from this CFA model were refined using the graded response theory model. This meticulous psychometric process aimed to enhance the precision and validity of the questionnaire’s items for a more robust evaluation of social participation among burn survivors. This was designed to develop measures precisely for different aspects of social involvement in the community.

### 5.4. Item Response Theory (IRT) and Differential Item Functioning Analysis (DIF)

Within the framework of IRT, specifically employing the graded response model, a thorough process was conducted to scrutinize the difference between observed and expected results of subjects in each category at the item level, utilizing Pearson’s chi-squared test [46]. Misfit items were discerned through an assessment of item discrimination parameters, guiding the decision on their inclusion in the refined model. To mitigate Type I errors, the Benjamini–Hochberg adjustment was applied. Responses varied based on gender, age, and employment status at the time of injury. In parallel, Differential Item Functioning (DIF) analysis was conducted to determine where individuals at the same estimated ability or function level respond differently to the same item based on external variables. DIF analysis, encompassing age, gender, race, and time since burn injury, was executed using a two-step IRT-based method, incorporating Wald chi-squared statistics to identify items exhibiting DIF [46,47]. DIF items were identified by differences across categorical groups in terms of ability levels based on selected demographic and clinical variables. This robust analytical approach aimed to unveil nuanced insights into item performance and ensure the validity and reliability of the refined model across diverse demographic and selected clinically related factors. Using these approaches, we found that the CAT developed was highly credible, reflecting excellent item performance and measurement properties for the model.

### 5.5. Reliability and Validity of the CAT

Several parameters are considered when evaluating a CAT application. The key parameters include reliability, validity, and useability. In this LIBRE study, we developed the CAT with 192 questions representing multiple social participation-related domains. This was administered to the same convenience sample of 601 burn survivors mentioned earlier. Demographic characteristics included 54.7% women with mean age of 44.6, average time since burn injury of 15.4 years, and average TBSA of 40%. The ultimate use of the confirmatory factor analysis ensured that the sample data fit the distribution from the given burn population. The LIBRE Profile ensured that there was a reference score for the individual scores earned by each participant so that the test was both accurate and reliable. The six domain-specific metrics were all standardized with a t-score distribution and a mean of 50 and a standard deviation of 10 based on the overall calibration sample of burn survivors. The final item bank following psychometric analysis yielded 126 items for the final calibrated item bank.

### 5.6. Simulation Studies

Before the CAT program was implemented, simulation studies were conducted to evaluate the testing parameters prior to live testing. This allowed confirmation that the CAT was functioning properly within the calibrated item bank. CAT simulations were run using the calibration sample to conduct real data CAT simulations; in this case, the CAT algorithm would select the item first with the highest information function for the mean score of the specific domain. The score and standard error were estimated using the weighted likelihood estimation. The optimal information was used to select the item and recalculate the score based on the subsequent response. Stopping rules required at least five items and no more than 10 with a reliability of 0.90. Therefore, while 60 items would be administered if all six LIBRE Short Forms are taken, the LIBRE CAT would produce scale scores administering between 30 and 60 items. The simulated CAT score and full item bank were compared for comparability for the range of the scores, ceiling and floor effects, and marginal reliability. The ratio was calculated between CAT and average full item bank responses, which then resulted in the time saved in the CAT simulation compared to the full item pool, with an assumed completion rate of three items a minute. The simulations validated well the CAT administration as compared to the full item bank being tested. Marginal reliability of the full item bank and CAT were 0.84 to 0.93, respectively [30]. Ceiling effects were less than 15% for all scales, and floor effects about 0%. The percentages of subjects with score reliability ranged from 72% to 91%. The average number of administered CAT items and number of items in each full item bank ranged from 0.22 to 0.47, and the average estimated time for CAT simulations was 1.94–2.63 min. Time saved with the CAT ranged from 3 to 7.25 min per scale for a domain compared to the full item bank being administered. The CAT demonstrated credible psychometric characteristics for future use by clinicians and patients in clinical practice.

### 5.7. LIBRE Fixed Short Form Profile

A short-form version of the LIBRE Profile CAT was developed, consisting of 10 items for each of the six domain scales selected from the large item pool based on high item discrimination, large score range coverage, adequate sample distribution, and a distribution of average item difficulty across each of the scales [48]. The LIBRE Fixed Short form Profile provides scale scores fairly comparable to the complete item bank metrics of the CAT. This is a useful substitute for the CAT when computer and internet access is not available. (The link to the LIBRE website is given at https://spauldingrehab.org/research/programs-labs/rehabilitation-outcomes-center/life-impact-burn-recovery-evaluation (accessed 4 March 2025)).

### 5.8. Testing of the LIBRE Profile Scores

The LIBRE Profile scores for the six domains of social participation in the community are each standardized using a t-score transformation with a mean of 50 and a standard deviation of 10. Prior work has provided important approaches for monitoring important guide posts for interpretation at the group level and individual subject level for determination of what is clinically and socially important and relevant. The LIBRE assessments can inform treatment plans and assist in program evaluation when compared with baseline values cross-sectionally and over time [49]. The profiles have been evaluated for test-retest reliability and convergent validity using other well-established metrics such as the YABOQ and PROMIS. The standard error of measurement (SEM) was also derived to provide the precision necessary for scores at the individual subject level. The minimal detectable change (MDC_90_ and MDC_95_) was established with the standard error of measurement (SEM) to quantify the precision of scores at the individual level [50]. This determined the lowest threshold for which change observed is not due to measurement error. The SEM scores were lowest for the Social Interactions scale, while those for the Work and Employment scale were the highest. MDC_90_ ranged from 6.08 to 7.86 points, while MDC_95_ ranged from 7.26 to 9.40 points across the six scales. The MDC values are important, as they provide the basis for what is deemed clinically relevant change for each of the six LIBRE domain scores.

## 6. Score Interpretation of the LIBRE Profile for Use in Clinical Practice

The interpretation of LIBRE Profile scores is important to promote use by clinicians in the clinical setting and burn survivors in the community. Prior work identified specific cut points for determining various levels of social participation and their importance [51]. Levels were established using a modified Delphi process consisting of anonymity, iteration, and feedback of results to a group comprising burn survivors, clinician experts, and psychometricians. A bookmarking process was used to review domain item maps, developed from calibration data, to establish score cut points that defined different levels of social participation. Using an iterative process means, standard deviations and the sample distributions were reviewed for each level. Level cut points with narrative descriptions were established for each domain, providing a context for understanding the meaning and interpretation of LIBRE Profile scores for each of the six domains. This work was useful for understanding the meaning of the numeric scores for use in clinical practice and by burn survivors in the community.

### 6.1. Analysis of the Clinical and Demographic Data Collected During the LIBRE Profile Field Testing

Clinical validity of the LIBRE Profile was established by querying the database obtained from the field testing of the item bank. This process described the data in a clinical context as one of the more-detailed datasets of long-term burn survivor social participation outcomes.

Social interactions and social activities were found to be long-term challenges for burn survivors [52]. Younger age and being married/living with a significant other were associated with higher scores on the Social Activities and Social Interactions LIBRE Profile scales. Individual item responses revealed that burn survivors scored lowest on items related to outdoor activities and feeling uncomfortable with their appearance.

Support group attendance was reported by 330 (55%) of 596 respondents who answered the question about peer support [24]. Support group attendees had larger burn sizes and were more likely to be >10 years from injury. Survivors who attended at least one support group scored significantly higher on three of the LIBRE Profile scales: Social Interactions, Social Activities, as well as Work and Employment. Overall, burn survivors who reported attending peer support group meetings had better social interaction scores than those who did not. This is the first reported association between peer support attendance and improvements in community reintegration in burn survivors [51].

Long-term differences in social reintegration outcomes were compared in burn survivors with and without work-related injuries [52]. Work-related injury data were examined based on data collected from 595 participants. Older participants (average age of 45 ± 16 years), those who were married (269 participants, 45%), and men (269 participants, 45%), were more likely to be burned at work.

After adjusting for demographic and clinical characteristics, burn survivors who were injured at work scored significantly lower (worse) on the Work and Employment domain.

Looking specifically at the Work and Employment scale of the LIBRE Profile, individuals who were burned at their workplace scored significantly worse on six of the 19 items within this scale. These individuals were sometimes more afraid to go to work and felt more limited in their ability to perform at work. Overall, burn survivors with work-related injuries reported worse work reintegration outcomes than those without work-related injuries. Identification of those at higher risk for work reintegration challenges, as well as a greater understanding of the barriers faced by these individuals when they do return to work, may enable survivors, providers, employers, and insurers to better use and target appropriate resources to enable optimal employment outcomes [24,53].

Work integration and retention after burn injury are key outcomes of social recovery. In a comparison study of these burn survivors and the general population, employed burn survivors scored higher on the Work and Employment scale of the LIBRE Profile, indicating higher participation [54].

Level of education has been correlated with overall health, health outcomes, and life expectancy, so another study aimed to determine whether level of education was associated with social recovery after burn injury [55]. Findings from the LIBRE Profile field test data suggest that burn injury during school-age years may have no impact on a survivor’s educational trajectory. The largest correlation between level of education and social reintegration was seen for the under-30 age cohort [56].

Significant differences in item response related to gender were found. Women were less likely to report being sexually active, with 40% of the female respondents reporting that they were not in a sexual relationship compared to 29% of the males [57,58]. Overall, men scored significantly better than women on four of the six LIBRE Profile scales (Sexual Relationships, Social Interactions, Work and Employment, and Romantic Relationships). After adjustment for demographic characteristics and burn size, men continued to score better than women in most of the areas measured by the LIBRE Profile. These gender differences are potentially important for managing individuals during the post-injury recovery period and provide a basis for further research. The likelihood of a burn survivor to be in a sexual or romantic relationship is not different from a population of 2000 adults without burn injuries [58].

Of the 597 burn survivors with complete data on age at injury, 165 (27.6%) sustained burn injuries as a child [59]. Those burned as children were more frequently female than those burned as adults (57% vs. 47%) and were also more frequently white and non-Hispanic (89% vs. 77%). Burn survivors who sustained injuries as a child fared at least as well as those burned as adults in a broad range of long-term social participation outcomes. The impact on long-term social participation outcomes of burn survivors was not significantly different between individuals with burns sustained during important developmental stages and those injured later in life. Future work can address how developmental stages relate to recovery in those with burn injuries, with some focus on social activities and resilience [58,59].

Burn size is an established clinical predictor of survival after burn injury. It is often a factor in guiding decisions surrounding early medical interventions. In examination of the LIBRE Profile calibration dataset, the relationship between burn size and community participation outcomes was complex [60,61,62]. Sociodemographic characteristics were compared between participants with small burns (≤40% TBSA burned) and large burns (>40% TBSA burned). Ordinary least squares regression models examined associations between burn size and LIBRE Profile scale scores with adjustments for sex, current work status, burns to critical areas, and time since burn injury. The analytic sample comprised 562 participants with burn size data. Forty-two percent of the respondents had large burns (>40% TBSA burned) and 58% reported smaller burns (TBSA ≤ 40%). In adjusted regression models, patients with large burns tended to score lower on the Social Activities and Work and Employment scales (*p* < 0.05) and higher on the Family and Friends scale (*p* < 0.05). Participants with burns > 40% TBSA scored lower for several individual items in the Social Activities scale and one item in the Work and Employment scale (*p* < 0.05). Increasing burn size was found to be negatively associated with selected items of Work and Employment and Social Activities, but positively associated with aspects of Family and Friend Relationships. Future longitudinal studies are necessary to assess and understand the long-term consequences of the social impact of burn injuries on adult populations [62].

### 6.2. Information from Prospective Use of LIBRE Profile for Clinical Queries

The LIBRE Profile was applied for assessment of the impact of adverse childhood experiences on long-term outcomes of adult burn survivors [63]. Adverse childhood experiences (ACEs), including child maltreatment and household dysfunction, define adverse events that occur before 18 years of age. National and state data report between 12.5% and 14.5% of the adult population report ≥ 4 ACEs (HIGH-ACE), respectively. HIGH-ACEs are associated with more chronic health problems. Using a new single-center population at the University of Iowa, inpatient and outpatient adult burn survivors (*n* = 53, ≥18 years of age, 66% male, burn size <10%, 32% flame, 67% visible burns, 2 weeks to 3 months from injury) were enrolled, completing the LIBRE Profile short forms.

Subjects also completed surveys assessing adverse experiences (ACEs-18), needs, strengths, and resiliency at consent, and pain, depression, post-traumatic stress disorder (PTSD), and social participation surveys at 2 weeks to 3 months postinjury. Demographics, burn, and hospital course data were also collected. Chi-square and student’s *t*-tests were used for descriptive analysis and to compare the groups (HIGH-ACE vs. LOW-ACE). The HIGH-ACE group (*n* = 24; 45.3%) reported more depressive symptoms (*p* < 0.04) than the LOW-ACE group (*n* = 29, 54.7%). HIGH-ACE patients were less resilient when facing stressful events (*p* ≤ 0.02) and more likely to screen positive for probable PTSD (*p* = 0.01) and to score lower on the LIBRE Profile, for the Family and Friends domain (*p* = 0.015). Findings from this exploratory study suggested that ACE screening may help detect persons with burn injury who are at risk for a more complicated recovery, thereby promoting personalized assistance in recovery.

## 7. Future Work

### 7.1. Trajectories Using the LIBRE Profile

Trajectories of recovery have been developed and validated for the burn populations to track the changes in outcomes using the six domains of the LIBRE Profile [59]. Results suggest that while there are improvements in social participation and community integration over time for the burn survivor, there are important limitations in their recovery as well. This suggests that burn survivors also suffer from a course not unlike a chronic condition over the longer term requiring resources and support services from the community. Trajectories on the basis of expected change from population data can assist the clinician in better understanding the courses of recovery for the burn survivor [62,64,65,66].

### 7.2. Development of Pediatric Parent-Reported Outcome Measures

Similar methods are underway to develop two separate instruments for parent-reported outcome measures for child burn survivors, including children 1 to 5 years of age (PreSchool LIBRE_1–5_), children 5 to 12 years of age (School-Aged LIBRE_5–12_), and a self-reported instrument for teenagers 12 to 19 years of age (Teen-Aged LIBRE_12–19_) [67,68,69]. The domains of interest are different between the age groups, and the complexity of the task of creating an instrument is increased by the need to consider the growth and development of the child. Efforts to link the progress of the burn survivor over time through the different age-specific versions of the LIBRE Profile will require dependence on anchor items and common metrics for key elements such as sleep, pain, and itch [70,71].

### 7.3. Translation of the LIBRE Profile into Other Languages

The LIBRE Profile fixed short form is currently being translated and undergoing testing for language and cultural validity in Mexican Spanish, Australian English, Taiwan Chinese, Turkish, and Japanese. Translations need to be sensitive to the cultural issues and differences in local dialects. The translations are conducted according to strict ISOQOL criteria, which include mixed-method approaches for both qualitative and quantitative research following Cosmin Guidelines [72,73]. Translations are well underway, and plans call for expansion of this program to other nations as well as regions within the US and Canada. Future comparisons are to be made across countries to assess important burn outcomes internationally.

### 7.4. Dissemination and Development of a LIBRE Application (APP)

The development of a mobile application (APP), LIBRE GO, for administration of the LIBRE Profile is well underway. This project is supported by funding from the App Factory (NIDILRR Grant #90DPHF0004), and LIBRE GO is currently available for download on the Apple Store (https://apps.apple.com/us/app/libre-go/id6670215871 (accessed 4 March 2025)) and Google Play (https://play.google.com/store/apps/details?id=org.mgb.libreapp&hl=en_US (accessed 4 March 2025)). The development process is a clinical-academic-community collaboration to translate the LIBRE Profile from a research product into a widely used application that meets the needs and preferences of stakeholders, including burn survivors, clinicians, and researchers. LIBRE GO uses state-of-the-art technology for assessing LIBRE scores at the “*n* of 1” level for purposes of profiling an individual’s trends over time for each of the LIBRE Profile domains [74]. A score report uses a graph and text to help users understand their scores and receive real-time benchmarked feedback for each domain. Additionally, LIBRE GO directly connects users with a wide range of resources from the Phoenix Society that empowers individuals to manage their own recovery. Integration of the Profile into the ‘Phoenix Society’, a national consumer organization of burn survivors, is planned for dissemination of this LIBRE GO mobile application across the country. Website and other Peer Support materials are currently in play.

### 7.5. The Efficacy of Clinical Interventions to Improve Social Participation

Clinically, the LIBRE Profile might be used by clinicians as a screening tool to select areas where a person might need additional support or would be a candidate for targeted clinical research. The LIBRE Profile can now be included as an endpoint in future clinical trials to assess the efficacy of a range of interventions designed to impact social participation of subjects being re-integrated back into their communities. The interventions include a range of future studies spanning impacts on the burn survivor focusing on family, social support systems, peer support groups, and community-based interventions. Last, realization for implementing and testing the effectiveness of new targeted personalized burn treatment interventions to improve social participation and quality of life after burn injury may be possible. Recognizing that an objective approach to implementing and evaluating the effectiveness of new targeted, personalized treatment regimens is feasible might enhance social participation and quality of life after burn injury.

The process of co-production of this instrument with clinicians, burn survivors, and their families laid out a detailed profile of symptoms/outcomes within domains that are most important to people living with burn injury. This information can be leveraged to design personalized care plans that include areas that specifically address burn outcomes. Further, utilizing the LIBRE Profile to evaluate programming and services within the burn community-based organizations as a self-management tool is being applied by consumer organizations such as the Phoenix Society.

## 8. Conclusions

The LIBRE Profile is a cutting edge, condition-specific, clinically relevant, psychometrically sound, reliable, and valid PROM. This assessment is useful to assess burn survivors’ long-term social participation outcomes, thereby informing our understanding of the needs of the population and targeting appropriate resources for burn survivors. It is useful for survivors to plan their lives and set expectations for future recovery. Combining PROMS and clinical data enables optimization of treatment interventions and provides a basis for personalized care [75,76]. The instrument can be used to guide care and spark discussion regarding areas of concern between the patient and their family and their health care provider.

## Figures and Tables

**Figure 1 ebj-06-00023-f001:**
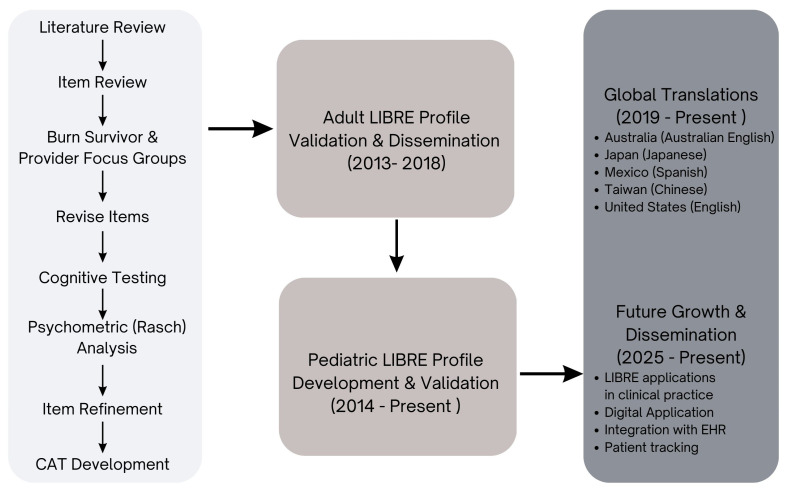
History of and future plans for LIBRE.

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
