# Peer review of "The Life Impact Burn Recovery Evaluation (LIBRE) Profile: Historical Overview and Future Directions"

_2673-1991, 2025, doi:10.3390/ebj6020023_

Round 1
Reviewer 1 Report
Comments and Suggestions for Authors
In this paper the development of the LIBRE Profile was described, in addition to psychometric evaluations and field testing. This work is important to disseminate in the burn field, since it could play an important role in the clinical practice and community of burn survivors. Social participation outcomes are linked to overall quality of life, and as shown by this research, consists of different domains, where burn survivors show unique profiles of scores.
I have some questions and suggestions:
- In line 84/85 the authors describe the importance of generic and condition specific PROM’s, and that ideally the two types are used together. It is unclear in the item bank that was used for the LIBRE profile, to what extend items are generic or burn-specific. It would be helpful to add this information.
- In line 150 ‘other clinical professionals from other fields’ are described. Please consider adding the types of professions.
- In line 156 the 192 question items are introduced, but it is unclear how these items were formulated, and based on what exactly.
- In line 232 reasons for challenges in return to work are described, but these reasons do not include reasons regarding work-related burn injuries (which is described later in the paper, but should also be part of this paragraph, in my opinion).
- In line 259 sexual behavior in burn survivors is described. A recent survey from the UK on this topic might be relevant: https://doi.org/10.1016/j.burns.2021.08.017
- The authors mention a sample of 601 burn survivors in line 298, but it is unclear if selection bias may have played a role in the recruitment and selection process.
- On the next page in line 351 the same n=601 is described, but it is not clear if this is indeed the same group.
- In line 368 the stopping rules do not include a clear description of the minimum and maximum total number of items in the LIBRE CAT (e.g. 30 and 60 items)
- In the part ‘LIBRE Fixed Short Form Profile’ which starts in line 383, it would be helpful to add the total number of items, which I believe is 60?
- In line 516 a new population is introduced from the University of Iowa, but details about this study are missing.
- The study results on ACE’s and LIBRE items could be emphasized in the abstract and conclusion of the manuscript, since these results should be replicated in other and larger samples.
- In line 543/544 a self-report instrument is introduced as part of developments currently underway. PROM’s can be completed by children starting at 8 years of age, so it would be interesting to see why 12 years of age is proposed.
- In line 597 the ‘long-term social participation outcomes’ are reported as result of the LIBRE Profile. I wonder if the authors can explain why the current profile is limited to social participation. Is there an initiative to incorporate mental and physical wellbeing as well? In most PROM’s that are used in the clinical and aftercare setting, the outcomes on all these domains are important to include and discuss.
Author Response
Reviewer 1: Minor
In this paper the development of the LIBRE Profile was described, in addition to psychometric
evaluations and field testing. This work is important to disseminate in the burn field, since it could play an important role in the clinical practice and community of burn survivors. Social participation outcomes are linked to overall quality of life, and as shown by this research, consists of different domains, where burn survivors show unique profiles of scores.
I have some questions and suggestions:
- In line 84/85 the authors describe the importance of generic and condition specific PROM’s,
and that ideally the two types are used together. It is unclear in the item bank that was used
for the LIBRE profile, to what extend items are generic or burn-specific. It would be helpful
to add this information.
The items were originally taken from both generic and burn specific assessments. They underwent considerable revision based on focus groups and cognitive interviews with burn survivors so that items were tested for their face validity. We have added some additional language in the text to reflect this. Ideally, generic and condition specific PROMs can be bridged to provide a wider understanding of impact on the life of burn survivors.
The following sentence was added into the paragraph mentioned by the reviewer (Lines 104-107):
. Items for the original item pool for the LIBRE Profile to be described were both generic and condition specific for burns. Based upon cognitive interviews and focus groups with burn survivors, generic items were modified so that they were condition specific.
- In line 150 ‘other clinical professionals from other fields’ are described. Please consider
adding the types of professions.
This point was clarified in the text (Lines 171-175)
In parallel, focus groups were conducted with 27 clinicians comprising 12 physicians including representatives from burn surgery, pediatrics, psychiatry, and physiatry and 15 other clinical professionals including nurse specialists in burn care, and representatives from several therapy divisions, psychology, social services and child life.
In line 156 the 192 question items are introduced, but it is unclear how these items were
formulated, and based on what exactly.
We thank the reviewer for pointing this out, and we agree that the method is unclear as written. This sentence was added to clarify this process.
Based on extensive literature reviews, consultations with clinicians specializing in burns, and focus groups noted above with direct input from burn survivors, 192 items were identified for inclusion in the field-testing survey. (lines 178-180).
In line 232 reasons for challenges in return to work are described, but these reasons do not
include reasons regarding work-related burn injuries (which is described later in the paper,
but should also be part of this paragraph, in my opinion).
The paragraph does include reasons for not returning to work among those with burn injuries and cites the works by Verma et al (#38). This work focuses on work related burn injuries. The following sentence was added:
The work by Verma et al. 38 indicates that return to work following a burn injury is related to social integration while not doing so is related to ones’ physical impairment and quality of life. (lines 257-259)
In line 259 sexual behavior in burn survivors is described. A recent survey from the UK on
this topic might be relevant: https://doi.org/10.1016/j.burns.2021.08.017
Thank you for this suggestion. At line 288 and after the following text was added, and references renumbered:
In a United Kingdom survey of burn units by Hurley et al.42 results are reported that sexual functioning is often not addressed by physicians in clinical treatment of burn survivors. (lines 290-292)
The citation for this is:
Hurley A, King I, Fiona MP, Baljit DS.
Addressing sexual function in adult burns victims: A multidisciplinary survey of current practice in UK burn units, Burns 48:4, 926-931 2022.
ISSN 0305-4179.
https://doi.org/10.1016/j.burns.2021.08.017
The authors mention a sample of 601 burn survivors in line 298, but it is unclear if selection
bias may have played a role in the recruitment and selection process.
The sample was a “convenience sample” and while selection bias may have played a role in the sample selection, it represented a rich range of clinical and sociodemographic characteristics. We have modified the sentence to read:
This rigorous and meticulous process involved the systematic administration of 192 questions spanning various facets of social participation to a substantial rich convenience sample of 601 burn survivors representing a range of clinical and sociodemographic characteristics. (lines 330-334)
On the next page in line 351 the same n=601 is described, but it is not clear if this is indeed
the same group.
The sentence has been modified as the following:
This was administered to the same convenience sample mentioned earlier of 601 burn survivors.” (lines 387-388)
In line 368 the stopping rules do not include a clear description of the minimum and
maximum total number of items in the LIBRE CAT (e.g. 30 and 60 items). • In the part ‘LIBRE Fixed Short Form Profile’ which starts in line 383, it would be helpful to
add the total number of items, which I believe is 60?
The following clarification was added to the manuscript:
Therefore, while 60 items are administered if all 6 LIBRE Short Forms are taken, the LIBRE CAT would produce scale scores administering between 30 and 60 items. (lines 407-409)
In line 516 a new population is introduced from the University of Iowa, but details about this
study are missing.
The following details are added to the text:
Using a new single center population at the University of Iowa, inpatient and outpatient adult burn survivors (n=53, ≥18 years of age, 66% male, burn size <10%, 32%flame, 67% visible burns, 2 weeks to 3 months from injury) were enrolled, completing the LIBRE-Profile short forms. (lines (557-560)
The study results on ACE’s and LIBRE items could be emphasized in the abstract and
conclusion of the manuscript, since these results should be replicated in other and larger
samples.
Thank you for this comment, however, the authors maintain that this is a paper about the development of the instrument and not about any clinical study. The ACE study contributes to face validity of the instrument and illustrates a potential use of the instrument. The discussion was amended to reflect that:
Clinically, the LIBRE Profile might be used by clinicians as a screening tool to select areas where a person might need additional support or would be a candidate for targeted clinical research. The LIBRE Profile can now be included as an endpoint in future clinical trials to assess the efficacy of a range of interventions designed to impact on social participation of subjects being re-integrated back into their communities. The interventions include a range of future studies spanning impacts on the burn survivor focusing on family, social support systems, peer support groups and community-based interventions. Last, realization for implementing and testing the effectiveness of new targeted personalized burn treatment interventions to improve social participation and quality of life after burn injury maybe possible. Recognizing that an objective approach to implementing and evaluating the effectiveness of new targeted, personalized treatment regimens is feasible might enhance social participation and quality of life after burn injury. Lines 624-626, and 633-636)
In line 543/544 a self-report instrument is introduced as part of developments currently
underway. PROM’s can be completed by children starting at 8 years of age, so it would be
interesting to see why 12 years of age is proposed.
During qualitative work for this overall pediatric LIBRE project, it was evident that the many burn outcomes relevant to this population were closely tied to the classic ages and stages of childhood. Currently the project is divided into 3 instruments, a 1–5-year-old, a 5–12-year-old (both parent-reported) and a 12-19 year old (person-reported). Within these age groups there is differential item functioning on some of the items within these age groups, resulting in a different set of short forms for different ages. Much work remains including a child reported version for the 8–12-year-olds and a parent reported version for the 12–17-year-olds. The teen instrument is still in the field-testing stage, the school age is seeking funding for validation studies, but the pre-school CAT is now built on a Qualtrics platform that we are in the final stages of testing. Focus groups are being assembled to attach meaning to scores. Translations, linking, domain and item enrichment, and increased dissemination all lay ahead.
In line 597 the ‘long-term social participation outcomes’ are reported as result of the LIBRE
Profile. I wonder if the authors can explain why the current profile is limited to social
participation. Is there an initiative to incorporate mental and physical wellbeing as well? In
most PROM’s that are used in the clinical and aftercare setting, the outcomes on all these
domains are important to include and discuss.
Social participation based upon the WHO ICF served as the conceptual basis for LIBRE focusing on two central concepts, societal role and personal relationships. The LIBRE instrument was designed to measure social functioning for the burn survivor’s recovery. Future work can consider burn specific physical function and emotional wellbeing.
Reviewer 2 Report
Comments and Suggestions for Authors
Summary
Thank you for submitting this paper for review. Firstly, the development and evolution of the LIBRE Profile as a burns-specific patient outcome measure (PROM) is an important contribution to the field of burns and this is its main strength. Attempts by the team to reduce the burden of responding and therefore increase the likelihood of people completing PROMS should be applauded and encouraged. However, I have struggled to understand the aim of this paper because it is unclear to me, and this is my key criticism of the paper.
Suggestions
Please situate this paper in context - it is supposed to be a historical overview and future directions, but this gets lost in the narrative and the content. On first reading I assumed that this was a brand-new tool and had not been published before. On reading the full article and exploring its references this is clearly not the case.
I would find it helpful to provide a specific timeline of the project and to clearly state the purpose of this paper, now. I think this is hinted at in the paragraph starting Line 87. If so, please make this clearer. LIBRE was developed some years ago but what is new about this paper now? This should be stated in the Abstract and expanded elsewhere. From the number of LIBRE papers cited in the references this is not the first time the burns community are hearing about its development and foundations or some of its findings. I have not read all the previous papers, but I would imagine there is a risk of duplicating material previously published and that should be avoided
Specific points of feedback and for clarification
Line 34 Is APP short for application or is it an acronym I do not know? If it is an app then surely app or application is more appropriate.
I would like some explanation as to why patients with large TBSA’s were used for the creation of the tool. A focus group (n=23) had a TBSA range of 18-94%, participants in the calibration study had a minimum of 5% TBSA and critical area involvement and participants in the CAT convenience sample had an average TBSA of 40%. Why? It almost appears as though those with burns of less than 5% have not been consulted and I would like to know why. Tt is well known that TBSA alone is a poor predictor of psychosocial morbidity. A brief search for epidemiological data led me to this 2015 paper analysing the (UK) IBID database of >80,000 patients where median burn surface area was 1.5% https://doi.org/10.1136/bmjopen-2014-006184. The burns community is made up of far more individuals with smaller burns than those consulted here. I would be concerned that prioritising the voices and experiences of large burns could have risked skewing the tool for the vast majority of burns survivors without needing to.
I was pleased to note that there is no mention of exclusionary criteria for those with acute mental health or psychiatric problems. If this is the case this is a strength of the inclusive nature of the development of the tool and I would deem that worthy of mention. So many studies exclude this important group within the burns population.
Line 139 Again, setting this in a time context is important. When was the literature review completed or last updated? If it was within the last five years there is no mention (nor anywhere in the paper or in references) of the CARe Burn Scales, Griffiths et al (2019) doi: 10.1093/jbcr/irz02 which could suggest that the literature review was not extensive. I would expect to see some mention of other tools that have emerged and especially those that might be in use in Europe as this is the EBJ.
Sec 4.1 It would be important to acknowledge and state the cultural context and specificity within which this study is situated, and the tool was created. I assume the USA. When measuring social impact and social reintegration post burn this should be located within specific social groups, communities and cultures. Even within English speaking countries like the US and the UK there will be differences in what social participation and community looks like. What were the cultural contexts and variation within the sample population? Culture is not even mentioned until Section 7.3 regarding translation. Language is different to culture.
Sec 5.5 I particularly applaud the attempt to reduce questionnaire burden by reducing the number of questions for participants without cost to the quality of the data produced. If anything this could be the cornerstone of the piece.
Line 390 Unfortunately the link to the site returns a 404 error which should be fixed prior to resubmission/publication
Line 452 This is really important and useful data that I have frequently observed clinically and has important implications for potential early intervention and support which would be different compared to those who did not sustain their injury at work.
Line 563 I checked, and Libre Go is not available on my Apple Store (UK) and I would be concerned it may only be available in the USA. I cannot test it for other countries. It cannot be said that it is available for download if this is geographically limited, this should be stated e.g. USA only. Given that this is a European Journal it would really be a shame if this is the case and weakens the value of the tool for Europeans. It would appear that right now non-US individuals can only use a physical version. That would also lessen the value of writing about the CAT version, which surely is a key strength of the measure?
I have a general question about when burns survivors were first involved in the study. As clinicians and researchers move to involve ‘subjects’ earlier and within the design phase of studies it would be good to know when the first point of involvement was.
As LIBRE has been around for some time now, is there data or inquiry into its actual uptake within burns services? How many people have downloaded the app? How many services include it within their routine or ad hoc measures? This would also be interesting to add and bring things more up to date.
I can only imagine how much work and effort has gone into this project. Thank you for your efforts so far.
Author Response
Reviewer 2: Minor
Summary
Thank you for submitting this paper for review. Firstly, the development and evolution of the LIBRE Profile as a burns-specific patient outcome measure (PROM) is an important contribution to the field of burns and this is its main strength. Attempts by the team to reduce the burden of responding and therefore increase the likelihood of people completing PROMS should be applauded and encouraged. However, I have struggled to understand the aim of this paper because it is unclear to me, and this is my key criticism of the paper.
Suggestions
Please situate this paper in context - it is supposed to be a historical overview and future directions, but this gets lost in the narrative and the content. On first reading I assumed that this was a brand-new tool and had not been published before. On reading the full article and exploring its references this is clearly not the case.
The authors wish to thank the reviewer for this comment and overall, for the thoughtful consideration of the work. The purpose of this LIBRE “Capstone” paper is to summarize the development and work on the LIBRE instrument to date to make the LIBRE Profile easier for people to use and understand. The dissemination of the instrument will hopefully be improved by providing a central place for information on the instrument and presenting a summary and integration of this body of work so that researchers can quickly direct any specific questions regarding its development, reliability and validity to a specific literature source. Additionally, assembling the body of work in one place demonstrates how the process of scientific co-production that includes the patients, clinicians and scientists is accomplished. This project began from identification of a problem, e.g. the lack of knowledge of the long-term health needs of burn survivors, which was brought to an advisory group by burn survivors and clinicians. This problem was addressed using co-production from grant writing through product development and continues through implementation. Inclusion of the stakeholders in the development of the product improves the uptake of the work. The Introduction was edited to reflect this by adding an introductory paragraph:
- Introduction
The purpose of this LIBRE “Capstone” paper is to summarize and integrate the first 10 years of the development and work of the adult LIBRE instrument and to make the LIBRE Profile easier for people to use and understand. The dissemination of the instrument through this paper provides a central place to find information on the instrument and present a summary of each paper so that researchers can quickly direct any specific questions regarding its development, reliability and validity to an original literature source. Additionally, assembling the body of work in one place demonstrates how the process of scientific co-production that includes the patients, clinicians and scientists is accomplished. This project began from identification of a problem, e.g. the lack of knowledge of the long-term health needs of burn survivors, which was brought to an advisory group by burn survivors and clinicians. This problem was addressed using co-production from grant writing through product development and continues through implementation. Inclusion of the stakeholders in the development of the product improves the uptake of the work. (line 44-59)
I would find it helpful to provide a specific timeline of the project and to clearly state the purpose of this paper, now. I think this is hinted at in the paragraph starting Line 87. If so, please make this clearer. LIBRE was developed some years ago but what is new about this paper now? This should be stated in the Abstract and expanded elsewhere.
We provide two figures, Figure 1 (History and Future Plans for LIBRE Profile) (after Lines 127 and 128 and Figure 2 (Time Line for LIBRE Project) after line 571. The first figure gives the development of the LIBRE Profile and its applications. The second figure gives the timeline for the LIBRE program past, present and future.
From the number of LIBRE papers cited in the references this is not the first time the burns community are hearing about its development and foundations or some of its findings. I have not read all the previous papers, but I would imagine there is a risk of duplicating material previously published and that should be avoided
The work is meant to be a summary of the history and integration of the development of the adult LIBRE Profile tool and an example of how an idea can flow from a clinical need to a deliverable. See response to first question, with an addition of a leading paragraph.
Specific points of feedback and for clarification
Line 34 Is APP short for application or is it an acronym I do not know? If it is an app then surely app or application is more appropriate.
APP is short for application, and this was corrected in the text. (lines 605-606)
I would like some explanation as to why patients with large TBSA’s were used for the creation of the tool. A focus group (n=23) had a TBSA range of 18-94%, participants in the calibration study had a minimum of 5% TBSA and critical area involvement and participants in the CAT convenience sample had an average TBSA of 40%. Why? It almost appears as though those with burns of less than 5% have not been consulted and I would like to know why. Tt is well known that TBSA alone is a poor predictor of psychosocial morbidity. A brief search for epidemiological data led me to this 2015 paper analysing the (UK) IBID database of >80,000 patients where median burn surface area was 1.5% https://doi.org/10.1136/bmjopen-2014-006184. The burns community is made up of far more individuals with smaller burns than those consulted here. I would be concerned that prioritising the voices and experiences of large burns could have risked skewing the tool for the vast majority of burns survivors without needing to.
While the burn survivors included in the focus groups were technically a convenience sample, they were recruited from a local peer support group where many of the individuals were highly involved in the burn community. The experiences of these people included their personal history; however, their knowledge also included the stories and interactions with many burn survivors which enriched our study.
Our objective was for the calibration sample to represent a wide range of rich clinical and demographics characteristics. To achieve this, we monitored these characteristics of the sample during recruitment to ensure that the sample mirrored the eventual users of the LIBRE Profile.
I was pleased to note that there is no mention of exclusionary criteria for those with acute mental
health or psychiatric problems. If this is the case this is a strength of the inclusive nature of the
development of the tool and I would deem that worthy of mention. So many studies exclude this
important group within the burns population.
Thank you for this comment. We have added a sentence to the text. (lines 336-338)
There were no exclusionary criteria for those with acute health or psychiatric problems for those with burns which enriched the samples heterogeneity.
Line 139 Again, setting this in a time context is important. When was the literature review completed or last updated? If it was within the last five years there is no mention (nor anywhere in the paper or in references) of the CARe Burn Scales, Griffiths et al (2019) doi: 0.1093/jbcr/irz02 which could3 suggest that the literature review was not extensive. I would expect to see some mention of other tools that have emerged and especially those that might be in use in Europe as this is the EBJ.
Thank you for this helpful comment. We have added a line in the Introduction (lines 68-67) in response along with the additional reference. The references have been renumbered.:
More recent PROMs specific to burns in CARe Burn Scale, published in 20186.
Griffiths C, Guest E, Pickles T, Hollén L, Grzeda M, White P, Tollow P, Harcourt D. The Development and Validation of the CARe Burn Scale-Adult Form: A Patient-Reported Outcome Measure (PROM) to Assess Quality of Life for Adults Living with a Burn Injury. J Burn Care Res. 2019 Apr 26;40(3):312-326. doi: 10.1093/jbcr/irz021. PMID: 30820556.
Sec 4.1 It would be important to acknowledge and state the cultural context and specificity within which this study is situated, and the tool was created. I assume the USA. When measuring social impact and social reintegration post burn this should be located within specific social groups, communities and cultures. Even within English speaking countries like the US and the UK there will be differences in what social participation and community looks like. What were the cultural contexts and variation within the sample population? Culture is not even mentioned until Section 7.3 regarding translation. Language is different to culture.
We agree with reviewer’s comment. Within section 7.3, cultural sensitivity and language nuances were addressed. Section 5.3 discusses the characteristics of the study group further, including the limitation of recruitment of only US and English-speaking Canadian subjects for the study.
Sec 5.5 I particularly applaud the attempt to reduce questionnaire burden by reducing the number of questions for participants without cost to the quality of the data produced. If anything this could be the cornerstone of the piece.
Thank you for this comment. The methodology described in the paper emphasizes the importance of reducing the respondent burden yet maintaining the rigor of the measure.
Line 390 Unfortunately the link to the site returns a 404 error which should be fixed prior to
resubmission/publication (lines 430-432)
The link is corrected. https://spauldingrehab.org/research/programs-labs/rehabilitation-outcomes-center/life-impact-burn-recovery-evaluation
Line 452 This is really important and useful data that I have frequently observed clinically and has important implications for potential early intervention and support which would be different
compared to those who did not sustain their injury at work.
The authors agree concerning this point. We think the yield of information beyond return to work is an example of the granularity of this condition specific instrument. The implications of this finding are discussed in the paper referenced:
- Schneider JC, Shie VL, Espinoza LF, Shapiro GD, Lee A, Acton A, Marino M, Jette A, Kazis LE, Ryan CM. Impact of Work-Related Burn Injury on Social Reintegration Outcomes: A Life Impact Burn Recovery Evaluation (LIBRE) Study. Arch Phys Med Rehabil. 2017 Nov 26.View Related Profiles. PMID: 29183751.
Line 563 I checked, and Libre Go is not available on my Apple Store (UK) and I would be concerned it may only be available in the USA. I cannot test it for other countries. It cannot be said that it is available for download if this is geographically limited, this should be stated e.g. USA only. Given that this is a European Journal it would really be a shame if this is the case and weakens the value of the tool for Europeans. It would appear that right now non-US individuals can only use a physical version. That would also lessen the value of writing about the CAT version, which surely is a key strength of the measure?
Wider access to the APP (Application) is on the docket. Administrative tasks involved in making the APP instrument available to European users is substantial but is planned pending future funding. Currently if the reviewer would like to see the APP, we can make that available.
I have a general question about when burns survivors were first involved in the study. As clinicians and researchers move to involve ‘subjects’ earlier and within the design phase of studies it would be good to know when the first point of involvement was.
Prior to writing the grant, the Phoenix Society for burn survivors had representatives on our advisory board and are true co-developers of the LIBRE-Profile. Ms. Amy Acton (Director of the Phoenix Society) and many other survivors across the US and Canada participated in every step of the development of this instrument. This is further demonstrated by the inclusion of Ms. Acton as a coauthor on this paper, representing the viewpoint of burn survivors. This point was made in the introductory paragraph.
As LIBRE has been around for some time now, is there data or inquiry into its actual uptake within burns services? How many people have downloaded the app? How many services include it within their routine or ad hoc measures? This would also be interesting to add and bring things more up to date.
There are a total of 64 individuals who have downloaded the LIBRE Profile from the Rehabilitation Outcomes Center Site (ROCS) website from 15 countries, including the US, Australia, Malta, Iran, India, Taiwan, Japan, Canada, the UK, France, Egypt, Kazakhstan, the Dominican Republic, Mexico, and Nepal. Of which, 29 individuals downloaded the LIBRE Profile for clinical use, 22 for research, and 13 for other use.
The LIBRE GO! App was soft launched in October 2024. A total of 110 individuals have downloaded the App. We are collaborating with the clinicians and community organizations to explore potential ways to disseminate this App and promote its use for supporting burn survivors’ recovery.
I can only imagine how much work and effort has gone into this project. Thank you for your efforts so far.
The authors greatly appreciate this comment. Thank you.
Reviewer 3 Report
Comments and Suggestions for Authors
The authors have compiled a review of their work to date on the adult LIBRE tool and described how it was developed, how it has been used and how it has led to more research questions. Although these have previously been published along the way, this review brings together the development and use of the tool thus far and also highlights the future direction.
I would support the publication for our readership and encourage collaborative research using the outcomes for burn survivors.
Author Response
The authors have compiled a review of their work to date on the adult LIBRE tool and described how it was developed, how it has been used and how it has led to more research questions. Although these have previously been published along the way, this review brings together the development and use of the tool thus far and also highlights the future direction.
I would support the publication for our readership and encourage collaborative research using the outcomes for burn survivors.
The authors wish to thank this reviewer and are grateful for his/her encouragement.
Reviewer 4 Report
Comments and Suggestions for Authors
The authors provide a well written paper and provide and nice overview of the LIBRE Profile’s development, psychometric foundations, and future directions and implementtion in clinical practices and burn survivor communities. I would like to congratulate the authors and the team with the excellent work that has been done with the LIBRE profile. The paper has a lot of information and perhaps the manuscript might be improved by removing some of the information to supplementary material. Especially in section/bullet 5. 'Advanced Psychometrics and CAT Development of the Adult LIBRE Profile' there a lot of information and some of it might be hard to comprehend for non- researchers/non-experienced readers. A suggestion could be to elaborate the figure in the manuscript 'Process for Development, Implementation and Dissemination of The LIBRE Profile' with some of the information in the text from item/bullet 5 and perhaps some infographics or nouns could help with describing theses aspects.
Author Response
Reviewer 4: Minor
The authors provide a well written paper and provide and nice overview of the LIBRE Profile’s
development, psychometric foundations, and future directions and implementation in clinical
practices and burn survivor communities. I would like to congratulate the authors and the team with the excellent work that has been done with the LIBRE profile. The paper has a lot of information and perhaps the manuscript might be improved by removing some of the information to supplementary material. Especially in section/bullet 5. 'Advanced Psychometrics and CAT Development of the Adult LIBRE Profile' there a lot of information and some of it might be hard to comprehend for non-researchers/non-experienced readers. A suggestion could be to elaborate the figure in the manuscript 'Process for Development, Implementation and Dissemination of The LIBRE Profile' with some of the information in the text from item/bullet 5 and perhaps some infographics or nouns could help with describing theses aspects.
We thank reviewer 4 for his/her positive comments. Regarding the Advanced Psychometrics section, we strongly recommend this section remains in the manuscript text. The rigor of the psychometrics is required for the credibility of the process for the instrument’s development. The methodology and the ability to deliver the LIBRE Profile as a CAT are some of the most important aspects of the instrument’s development and demonstrate the strong foundation of the survey.
We have added greater specificity for the first figure (edited Figure 1: “History and Future Plans for LIBRE Profile” (to follow line 127) and for a second figure, Figure 2: “Timeline for LIBRE Project” to be placed after the Subtitle “Future Work” (line 574) with details regarding the timeline for the development and dissemination of the LIBRE.